# Disconnection and the Healing Practice of Imagination for Mormon Environmental Ethics

Kristen Blair

School of Theology, Boston University, Boston, MA 02215, USA; kblair18@bu.edu

**Abstract:** The Church of Jesus Christ of Latter-day Saints possesses a subversive and fecund interpretation of the Christian creation narrative. This interpretation, denying creation *ex nihilo*, bespeaks a particular attention to and care for the living earth. However, Latter-day Saint praxis is wounded by a searing disconnect between the theopoetics of its conceptual creation and its lived practice. I argue that the Church must understand this disconnect as a wound and attend to it as such. I turn to theopoetics, arguing that it is in the lived practices of Latter-day Saints engaging somatically with the Earth that can restore our imaginative potential and move toward healing. I begin by exploring the Christian conception of creation *ex nihilo* and juxtapose this with the Latter-day Saint understanding of *formare ex materia*. I then explore the implications of such a cosmology for environmental ethics and probe the disconnections between theory and practice in Mormonism broadly construed. I propose that the healing salve for disconnection is imagination, a salve found in the first heartbeat of the Latter-day Saint story. I speak with Latter-day Saint theopoetics and indigenous voices, proposing ultimately that is with them that the healing of theology and praxis must begin.

**Keywords:** Church of Jesus Christ of Latter-day Saints; theopoetics; environmental ethics; creation; ex-nihilo; theological healing





The Church of Jesus Christ of Latter-day Saints (colloquially known as the Mormon church[1] is not known for its environmental ethic. This is surprising when one becomes even passingly acquainted with Mormon doctrine, which teaches of an animate earth in which all existing things possess eternal spirits. Indeed, the Mormon conception of creation sings with prophetic imagination; Joseph Smith, the prophet of Mormon restoration, reinterprets the biblical creation story to embrace chaos and mystery, femininity and shadow. His reclamation rains with subversive reinterpretation and sends roots into the eager soil of a Christian restoration and new beginning.

Mormonism's denial of creation *ex nihilo* and its restoration of creation out of chaos is the cosmological and, indeed, theological foundation of a religion so connected to earth and materiality as to be inseparable. The Mormon doctrine of the creation implies an ethic of care and respect for all material life, yet there is no widely adhered to or well-known environmental ethic in Mormon practice. I interpret this as a deep wound in the body of Mormonism, a gaping and perplexing disconnect between doctrine and practice that harms the collective body. I attend to some of the potential sources of this wounding to locate its continued festering in the landscape of disrepair in which its brokenness lies to discern how to bridge doctrine and practice (Walton 2020). I contend that if the Mormon people are to embrace the environmental ethic suggested in the creation narrative, Saints[2] must reclaim the untamed powers of imagination. The Mormon story's first placenta was prophetic imagination, and it is this imagination I search for in attending to the wound of disconnection. I seek to re-member the broken Mormon body and awaken her with the breath of imagination. To inspire imagination, I search for a Mormon theopoetic of creation with power to heal the wounds of disconnection and revitalize the animation found in the people's first imaginings.

## 1. Creation *Ex nihilo*, Creation *Ex profundis*, and *Formare Ex materia*

### 1.1. Creation Ex nihilo

Cosmology shapes ethics, dogma, and eschatology. Investigating the roots of Christian cosmology reveals the mouths of the theological rivers flooding our religious imaginations. Accordingly, I must begin at the beginning. The fourth century Nicene-Constantinopolitan Creed firmly establishes the position which will come to characterize Christian cosmology: creation *ex nihilo*, out of nothing. The dogma established in the creed addresses heresies including Gnosticism, Marcionism, and Arianism (Norman 1977) that would be deemed anathema to Christianity. Creation *ex nihilo* responds not just to gnostic ideas about the world, but to Greek philosophical understandings of matter, possibly in defense of Christian material resurrection (Hubler 1955)[3] Creation *ex nihilo* first emerges in the second century with apologists Tatian and Theophilus of Antioch and is later developed by Augustine and Aquinas.[4] It is made doctrinally binding in the Nicene Creed, which asserts with authority that God had no precedent and that creation was entirely sculpted according to His prerogative.[5]

Before the binding, however, creation had more breathing room. According to Catherine Keller, "[u]ntil the late second century, Jewish and Christian interpreters seem to have assumed that the Creator formed the creation from some depersonalized version of . . . primordial stuff" (Keller 2003, p. 15). Justin Martyr's second-century *Apology* states, "we have been taught that He in the beginning did of His goodness, for man's sake, create all things out of *unformed matter*," (Martyr 155, chapter 10, emphasis added), a statement much easier to harmonize with Greek conceptions of being and non-being than the later postulation of creation out of nothing. Justin's view is also in-line with Jewish interpretation of the Hebrew book of Genesis, which does not suggest creation *ex nihilo*. The patristic tension is philosophical; with pressure from formidable opponents, the fathers sought to defend a coherent theology against accusations of polytheism and create a cogent response to philosophical debates, such as the problem of evil. Creation *ex nihilo*, though not explicitly scripturally based, is intimately connected to the developing conception of the Godhead that continues to characterize the Christian tradition (Norman 1977, p. 318): God had to have created out of nothing in order to protect His omnipotence and omniscience. To encapsulate God's nature, the fathers subsumed the mystery of origin.

### 1.2. Creation Ex profundis

Constructive Theologian Catherine Keller probes creation *ex nihilo* in *Face of the Deep: A Theology of Becoming*. She traces the history and development of this doctrine so crucial to Christianity and points to deeper concerns necessitating the contained creation formed in an *ex nihilo* doctrine. The dark, the void, the primordial chaos suggested in the Genesis 1 account of creation stands in playful contrast with the Genesis 2 Babylonian epic, and in stark contrast with an *ex nihilo* narrative, which drowns the text's voice by third century AD. But beginning, Keller argues, is not origin. *Bereshit*, 'in beginning,' is not strictly linear or implicative of first causation. *Bereshit* signifies a start, a signifier on time (Keller 2003, p. 9). In the beginning, then, "the earth was *tohu vabohu*, and darkness was upon the face of *tehom* and the *ruach elohim* was vibrating upon the face of the *mayim* . . . ." (Keller 2003, p. 9). Keller interrogates the Hebrew, locating the pulsing heartbeat from within an ancient world, and troubles the waters of patristic interpretation. Before *bereshit* there was *tehom*. *Tehom*, Keller argues, has faces and being. She is the chaotic darkness so threatening to the Omnipotence that will come to reign over all the earth. "Genesis 1," Keller insists, "betrays no fear of the dark, no demonization of the deep, of the sea, its she and its dragons" (Keller 2003, p. 30).

The oceanic immensity prevalent in Babylonian myth, the chaos, the monstrous, the dark, the decidedly female and sexual, this is *tehom*. *Tehomophobia*, the fear of this immensity of the unknown and uncontrolled, "dot the Bible." If tehom is understood as evil, Levenson's critique is critically important to a robust theodicy. But, Keller progresses a theology of creation *ex profundis*, out of chaos, reading "the *tehom* not as the evil, but as the active potentiality for both good and evil" (Keller 2003, p. 92). The demonization of darkness, she argues, "belongs to the foundational *tehomophobia* of Christian civiliza-

tion" (Keller 2003, p. 200). Keller's *ex profundis* creation motivates a pedagogical impulse to "move more gracefully in the dark," (Keller 2003, p. 212) pointing out a particularly unpleasant implication of creation ex nihilo: in elevating God's omnipotence, creation ex nihilo also denies (or at least softens) human responsibility for the world (Keller 2003, p. 212). Creation *ex profundis*, on the other hand, "requires our entire participation" (Keller 2003, p. 214).

### 1.3. Formare Ex materia

Keller's troubling of creation *ex nihilo's* waters flows into multiple streams of Christian thought. Many Christian theologians recognize the troubling implications of creation *ex nihilo* (Jennings 2010). The *bereshit* of the biblical story is a non-trivial consequence to all that will follow in the narrating of the human condition. It matters, in other words, where and how the story begins. A Christian tradition that does not employ this dogma in its cosmology, then, has the potential to walk refreshingly new paths. The Mormon Church contains a unique conception of creation, one that explicitly denies creation *ex nihilo*.

In addition to translating ancient texts by revelation, Joseph Smith undertook a translation of the Hebrew Bible as part of his prophetic mission. Smith was particularly troubled by the widely distributed translations of Genesis in his day. He wrote:

> The word 'create' came from the word *bāra'*, which does not mean to create out of nothing; it means to organize, the same as a man would organize materials and build a ship. Hence, we infer that God had materials to organize the world out of chaos—chaotic matter, which is element and in which dwells all the glory. Element had an existence from the time He had. The pure principles of element are principles which can never be destroyed; they may be organized and reorganized, but not destroyed. They had no beginning and can have no end. (Smith 1971)

Expanding on this distinction, and as part of his translation of the Hebrew Bible, Smith articulated the book of Moses as a prequel to Genesis. The book of Moses situates the cosmology Smith claims to be restoring.[6] In the first chapter, Moses meets God face to face. In the second and third chapter, Moses sees the creation of the world in vision. The sequence parallels Genesis with a few deviations. Following the vision, God tells Moses: "every plant of the field before it was in the earth, and every herb of the field before it grew. For I, the Lord God, created all things, of which I have spoken, spiritually, before they were naturally upon the face of the earth."[7] This concept, along with the deliberate translation of Genesis from *bāra'* as 'create' to *bāra'* as 'organize,' shapes Mormon cosmology.

Smith posits two stages of creation, one spiritual and one physical. Juxtaposed with Augustine's two-stage creation, in which the first stage is God's creation of matter[8], Mormon theology posits a spiritual existence that is uncreated by and coeternal with God. Smith records God's speaking:

> "For by the power of my Spirit created I them; yea, all things both spiritual and temporal. First spiritual, secondly temporal, which is the beginning of my work; and again, first temporal, and secondly spiritual, which is the last of my work—Speaking unto you that you may naturally understand; but unto myself my works have no end, neither beginning . . . Wherefore, verily I say unto you that all things unto me are spiritual, and not at any time have I given unto you a law which was temporal" (my emphasis added). (D&C 29:33–34)

Adherents of Smith's translations[9] take literally the idea that God's work is without beginning or end. Creation was *a* beginning, but not *the* beginning. The creation of our world, according to Latter-day Saint cosmology, occurred from unorganized matter, from dark, void, and chaos. Creation was an organization: *Formare ex materia*[10] or, less pretentiously, organizing existing material. In other passages of scripture and doctrinal teaching, Smith and other leaders teach that our human spirits (intelligences) are eternal, uncreated by God: humankind, in the form of intelligences, were "also in the beginning with God."[11] "Intelligence, or the light of truth, was not created or made, and neither indeed

can be."[12] Indeed, another apocryphal text revealed to Smith taught of a pre-existence, wherein intelligences communed with Jesus Christ and collaboratively participated in the creation of a world: "And there stood one among them that was like unto God, and he said unto those who were with him: We will go down, for there is space there, and we will take of these materials, and we will make an earth whereon these may dwell."[13]

Mormon teachings[14] imagine a creation headed by Jesus Christ as an intelligence, aided by the intelligences that would become mortal beings. Moreover, the material from which all life was organized is inherently spiritual, eternal, with no beginning and no end. All life found on the earth is thus spiritual at its core, possessing uncreated, recycled, and organized material with a marvelous spiritual history.[15] For Mormons, God did not simply engage in a creative frenzy over a select time-period, creating like an artist the good and the evil according to God's will and pleasure. God, with the aid of other intelligences, organized material and set it into evolutionary motion.

This is prophetic imagination: an "alternative to the consciousness and perception of the dominant culture (Brueggemann and Hankins 2018)." Centuries before Keller and constructive theologians arguing in the same vein,[16] Smith imagined that creation is not *ex nihilo*. It is out of chaos, *tehom*, "the chaos that cannot be controlled, the mystery that cannot be solved" (Keller 2003, p. 23) Smith also precedes Keller in imagining that creation from chaos clarifies the nature of God. For Keller, creation *ex profundis* was a real threat to the work of the early church fathers. They sought to protect the image of God's perfect omnipotence and omniscience. But Keller points to *ex profundis* as giving rise to "the destructive chaos in the universe" as well as "the beauties of complex order" (Keller 2003, p. 92). She takes up theodicy, articulating the role of a creation narrative in ideas about the origins of evil. Mormon understandings, likewise, take from the creation narrative grounding concepts about God's nature. God is not the author of good or evil; God sets into motion. This does not birth the Deist imagining of God as clockmaker; Latter-day Saint teachings suggest a personal and present God that weeps for human suffering and woundedness.[17] "This is my work and my glory," God tells Moses in Smith's translation, "to bring to pass the immortality and eternal life of [humankind]" (D&C 29:32). The implications of this cosmological imagination are profound, winding their way into theodicy, eschatology, and ethics. In what follows, I focus on the implications of the Mormon creation narrative for a clear environmental ethic.

## 2. Locating a Mormon Environmental Ethic: Lost and Found

Returning to the revelation of Joseph Smith, I track the divine word about the order of creation: "first spiritual, secondly temporal which is the beginning; and again, first temporal, and secondly spiritual, which is the last of my work" (D&C 93:29 and Moses 1:39). I have traced the Mormon conception of "beginning" already, not as an out-of-nothing origin, but as a place of beginning agreed to by an assemblage of intelligences. The spiritual, therefore, preceded the physical. Intelligence—the spiritual matter which is neither created nor made—undergirds all of creation as we engage it. The tree outside my window as I write is first spiritual, eternally existent beyond my perception of its branches tasting the spring air. But the same tree is clothed in "temporal" material. It is matter—I can touch it, watch it move gracefully with the wind. My one-year-old can finger the blossoms stretching hopefully in the sunshine. It is physical, rooted in the material world.

This, I argue, is the heart of Smith's revelation: while the spiritual nature of all things preceded all material existence and will succeed all temporal death and decay, the order of earth's creation minds the material. To the tree, and to my beholding of the tree, the divine speaks: "first temporal, and secondly spiritual, which is the last of my work . . . for this is my work and my glory, to bring to pass the immortality and eternal life of [humankind]."[18] God's work so declared is to redeem the physical world and those inhabiting it. Not to bring to pass the immortality and eternal life of intelligences, or of spirit, but of human beings and our spiritual essence and physical embodiment, so intertwined as to be inseparable.



The Mormon understanding of creation welds the spiritual and physical in such a way as to be indivisible, melded toward the vision of salvation in which Earth itself is our eternal home.[19] If the physical (temporal) is preciously and permanently connected to the spiritual, surely it follows that one cannot favor care for one over the other. It is a sad misunderstanding, in other words, to imagine that one's "eternal soul" is an entirely separate matter from one's body. Moreover, if as Smith imagined the earth and all that lives upon it is animate, possessing spiritual life which existed eternally prior to mortality, human engagement with the world matters (to say nothing of the falsity of anthropocentrism). In short, a connection between theological understanding and environmental ethics seems a natural one by this reading. Yet, the body of Mormonism is wounded by a sharp disconnection between belief and practice.

To imagine Mormonism embodied prioritizes the actual bodies of believers. A collective Mormon body includes black bodies and brown bodies, disabled bodies and gendered bodies—it is a body in the gathering of bodies. The Mormon church's emphasis on gathering as well as its engagement with Paul's body of Christ rhetoric justify this imagination. In imagining Mormonism as a body, I invoke an awareness of the tender fragility of bodies generally. Bodies can be sick, they can be glad, and they can be wounded. Wounding, as I see it, is a metaphor for that which is accumulated and hidden away in shame, pain, or fear.

I locate the Mormon body's woundedness in its activity of disconnection. The Church's official website contains a page devoted to "Environmental Stewardship and Conservation," which teaches that "The earth and all things on it are part of God's plan for the redemption of His children and should be used responsibly to sustain the human family ("Environmental Stewardship and Conservation"). However, this resource is referenced in only one official teaching manual. It does not appear in any formal handbook or address to the general church. Mormon communities have two basic pedagogical systems: weekly Sunday meetings, and bi-annual general conferences, the teachings from which are generally the topic of discussion at the weekly Sunday meetings. Accordingly, if neither directly engages the topic of Environmental Stewardship and Conservation, the topic is unlikely to be accessed, taught, or discussed on a regular basis in most Mormon communities. This relatively unknown and underutilized resource could be utilized much more meaningfully if it were the product of a robust dialogue with its members.

This dialogue is vital, particularly given the ideological predisposition of Church members in the twenty-first century. Recent surveys indicate that Mormons on the whole continue to align with conservative policies and often resist environmental protections (Ward 2012), and Utah politicians' anti-environmentalism stances are frequently associated with the Church by non-members and members alike, despite the growing population of Mormons outside of Utah (Handley 2001). Indeed, Mormon ecologist Steven L. Peck notes that in the last thirty years, "Utah's representatives had among the poorest records of environmental protection in the nation" (Peck 2006). He explores Mormonism's complicated relationship to science, noting the "profound love of nature" among many Mormons which is perplexingly accompanied in the same people by vehement climate change denial and denunciation of environmental legislation (Peck 2006, p. 175). He traces this disconnect to a skepticism about scientific, rather than religious ways of knowing, as though the two are in opposition, a well-known divide in the history of Christianity generally. William Rudy similarly traces the "environment-religious schism" that has dominated Christianity and seeped into many representative strands of Mormonism despite distinctive doctrines keeping the church and many of its members disconnected from collective discussion about environmental ethics arising from the faith (Rudy 2006). The continued alignment of Mormonism with political and social conservatism when it comes to environmental ethics is a tragedy, but I also interpret it as a wound.

Indigenous peoples, many of whom have known and respected the spiritual nature of all life far longer than Mormons,[20] teach that the animation—even the personhood—of all beings begs human respect (Kimmerer 2013). The Honorable Harvest, practiced by some

Indigenous peoples as described by Robin Wall Kimmerer, sets up an ethic based on this knowing of the earth:

> Know the ways of the [plants] who take care of you, so that you may take care of them.
> Introduce yourself. Be accountable as the one who comes asking for life. Ask permission before taking. Abide by the answer.
> Never take the first. Never take the last. Take only what you need.
> Take only that which is given.
> Never take more than half. Leave some for others. Harvest in a way that minimizes harm.
> Use it respectfully. Never waste what you have taken. Share.
> Give thanks for what you have been given.
> Give a gift, in reciprocity for what you have taken.
> Sustain the ones who sustain you and the earth will last forever. (Kimmerer 2013, p. 183)

This kind of ethical standard is based on an understanding that nature lives and has being. Indigenous teachings, such as the Honorable Harvest, move from belief to practice with profound precision and integrity and thus honor belief. Mormon teaching of the spirituality of nature, the eternal animacy of all that lives on this earth, and the radically material vision for heaven which the tradition has claimed begs for practices which honor these truths and give them life. If it is true that the order of creation Mormons espouse minds the physical, what must the tradition do accordingly? If it is true that all life possesses eternal spirit, how must members live respectfully? Throughout Mormon history, notable Mormon academics have taken up these questions in profound ways.[21] Their pleas, however, appear to have gone largely unheeded.

In short, despite the attempts of Mormon academics, the Mormon people considered generally are not communally engaging the questions necessary to honor the shared belief system. Beliefs such as those I have rehearsed are poised to alter the way Mormons think about consumption, conservation, and recreation, yet I have presented evidence suggesting that Mormons on the whole remain resistant to environmental ethics. Indigenous traditions recognize that belief without responsive action lacks integrity. When people of faith fail to interrogate our beliefs and question how they lead us to live, when we lack responsive action to faith, we abandon the integrity that companions an ethical life. This large-scale disconnection between faith and practice is a wound in the Mormon body. Continuing to move this collective body in the world without attending to this disconnection is like running on broken legs; the running continues to harm the body with its polarized approach to environmental issues, resisting dialogue and meditation on the connection between belief and practice. In order to heal, Latter-day Saints must understand how they have been dismembered.

Like all traditions, Mormons are not monolithic. There are many possible reasons for the disconnection I have outlined between Mormon doctrine and Mormon practice, and the disconnection does not necessarily encompass all Church members.[22] Patrick Mason suggests that Mormon conservatism is a byproduct of the "very real persecution they had received in the nineteenth century." In response, he argues, "our people . . . metaphorically built a fortress church in order to protect ourselves and our precious holdings from invaders (Mason 2020, p. 6)." If Mason is correct, a Mormon history of persecution led to a Mormon culture of retreat and fortressing—disconnection from the world, literally. Mormon sacred space became a signal for safety, for respite from the world. Temples, the holiest houses of worship within Mormon culture, came to represent heaven removed from earth.

Jason Brown makes the case for Mormonism's retreat into political conservatism as the source of the disconnection, pointing to the "deep polarization of environmental issues on the American political landscape during the last fifty years." He argues that "when those who would advocate for environmental issues become stereotyped with free love, drug culture, and secularism, conservative Mormons tend to stop listening (Brown 2011, p. 69)." This perspective, aligning with Peck and Rudy, begins to explain how political affiliation

could become tantamount with religious belief. Similarly, Terryl and Fiona Givens argue that despite Mormonism's hope to restore the keys to Christ's promise to make "all things new," (Givens and Givens 2020) we are "in some ways ... still living and believing according to paradigms of the past" (Givens and Givens 2020, p. 3), adhering religiously to common dogmas established with Reformation and Counter-Reformation thought. From this reading, Joseph Smith's interpretation of the biblical creation story exemplifies the prophetic imagination. But, over time for safety and protection, we as a people borrowed dominant cultural imaginations, including the schism between environment and religion.

However the Church got here, so wounded, they must reckon with the fact that Joseph Smith imagined the *bereshit* of humankind in new ways, certainly in contradiction to the dominant imagination of the nineteenth century. His revelatory methods—somewhat magical, certainly unorthodox, almost entirely devoid of official theological or other training, and influenced by anything he personally found resonant—give rise to a creation story demanding responsible attention to the natural world. In order to fit into the dominant culture of Mormonism's birth-time, perhaps the Saints as a people have neglected the prophetic imagination of Joseph Smith. By overemphasizing the spiritual as though it is separate from the physical, Latter-day Saints deny the material, time-bound physicality of creation. Saints deny the unity of creation and the imperative of God's work and glory in which we participate, attempting to separate what was never intended to be parted. Core Mormon teachings glistening from the tradition's conception emphasizes the significance of a material creation that gives physicality to spiritual intelligence. Mormon eschatology does not separate the two—they are united eternally.

In locating "original" versus "contemporary" ethics and practices, I am engaging in a game of hide and seek, lost and found. I find the steady heartbeat of a clear environmental ethic in the original prophetic imagination of the Mormon creation story. I seek this ethic in the dominant cultural imaginations Saints have borrowed over time and employ in worship today, and I return empty-handed. Saints have broadly failed to locate the communal power to imagine a living creation and act accordingly. Yet, with grace, the same Saints remain in the ruins of this disconnection, this gaping wound piercing the collective Mormon body. A re-membering of the body, I argue, can occur only by reviving the practices that first gave our ancestors spiritual life. Mormons can re-member by stepping once again into the vulnerable, un-fortressed space of imagination.

## 3. Reviving Communal Imagination: Mormon Theopoetics of Creation

By inviting an orientation to the imaginary and creative, I am calling for a Mormon awakening to the realities of the first transformative beliefs and the wounding we are living with. This is of course only the beginning; Latter-day Saints must awake and arise to practical engagement with the earth. My suggestion is, however, that Saints can only acknowledge a collective wounding and engage as necessary by returning to imagination as a spiritual and healing practice. Reviving the powers of imagination on a communal level, re-membering, requires new kinds of engagement. Joseph Smith imagined radically, magically, and otherwise.[23] Mormonism derived from this prophetic imagination and was sustained by the imaginations of devoted followers willing to live and die for the vision they shared. Thus, Mormons are equipped in the very foundations, origin stories, and teachings, to practice imagining in a radically democratic fashion, to imagine as a spiritual practice of healing division and refocusing attention. To revive imagination and begin to heal our dismemberment, I turn to theopoetics.

Heather Walton writes, "'Theopoetics' is the term used by a vibrant and diverse movement of artists, activists, and scholars who fully accept the brokenness of contemporary theology—indeed, who take this as their starting point" (Walton 2020, p. 161). Walton's vision of theopoetics is "a project in ruins. Sacred speech that touches earth and employs mud language"(Walton 2020, p. 161). If the lack of Mormonism's general response to environmental issues is indeed a brokenness and wounding, as I have suggested, theopoetics is a quest for speech that seeks its first mother, imagination, and not its adoptive parents (in this

case, environmental conservatism) ([Walton 2020](), p. 168). As a healing practice, theopoetics turns inward, to our embodied and situated locations, using non-dominant language to bring attention to that which is easily overlooked or misunderstood. To recover the creativity to imagine otherwise, to reject that which holds no water, requires a return to the river bottoms of imagination. Theopoetics searches for language to capture embodied experience rooted in place and context. It locates artistic engagement in non-dominant language, like art and poetry, as sites of theological reflection in themselves.[24] Theopoetic engagement asks us to pay attention to the places and experiences that might otherwise be overlooked. With that attention, theopoetics opens avenue for reflection and questioning, for imagining and relating from a place of personhood, love, and knowing. Theopoetics bids us notice. Receive. Ask. Respond, identify, and revive imagination. Consider the following examples of imagination drawn from various embodied response to the land. I begin with my own imagination and then trace the profound potential of a few theopoetic voices within Mormonism.

*Toward a Mormon Theopoetic of Creation*

When I go home to Provo, Utah, I make my way to Rock Canyon as soon as I can. I long for Rock Canyon, dreaming of the mountains when I am far from home. My body remembers the feel of the canyon's pathways; when my feet are once again pointed toward the mountains, I can picture every turn of the trail. I know when the curve of the rocky path will open to water and which sloping hills to anticipate after every bend. I hope these mountains will see me grow old. I would like them to receive my frail body when I am quiet on this earth, knowing that of all the places I have been, the mountain has known me the longest. Rock Canyon has been audience to my most vulnerable conversations, inhibitions lowered in the certainty of my moving feet. In Rock Canyon I discover prayer when words run dry and the rushing waters continue. In Rock Canyon I encounter the silence of grief, and the space for it. There is no heaven when I go to Rock Canyon. Just earth, and my ragged breath, and the day.

Rock Canyon is the place of my first remembering, my re-awakening to the embrace of the earth. It is my greatest teacher, wordlessly connecting the religious doctrines I was raised with to the way my feet moved on the mountain. In Rock Canyon, it is easy to believe the doctrines of the Restoration. The spirituality of the trees, the rocks, the air above the inversion, is obvious. On the trails, I imagine my response to the grace of creation.

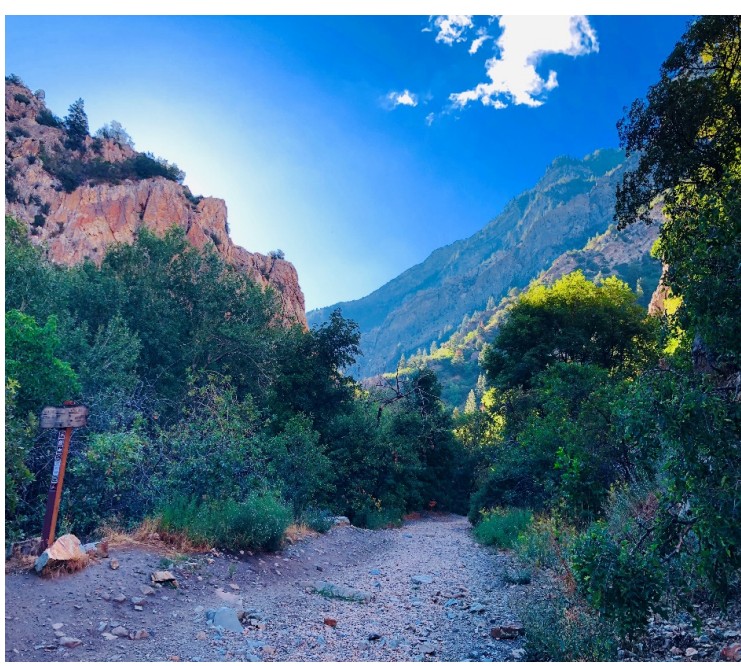

Rock Canyon, Provo, UT, July 2019. Image credit: author.

For George Handley, a scholar and poet whose work has awakened attention for Mormonism's underutilized environmental ethics, a place of imagination is the Provo River. His book, *Home Waters: A Year of Recompenses on the Provo River*, is situated between polemic and apology, a poetic meditation engaging ecology and faith, life, and death. "Scarce beauty," he writes, "is a gift, not a right. It merits love, not lamentation; love enough to make recreation a re-creation, a way of becoming unfamiliar again with the world, of working to blur the horizon line between heaven and earth (Handley 2010, p. 189)." For Handley, that work of blurring is intimately connected to his Mormon faith, which offers "hope that mundane, physical life, when properly cared for, might become the stuff of eternity" (Handley 2010, p. 121). The "work" he is proposing is both practical, calling readers toward ethical environmental living, and metaphysical, searching for language with which to recover the "fundamental relationship" (Handley 2010, p. 14) between humans and earth. He returns to the poetry of grace, Mormon theology, and poetics—theopoetics. From the Provo River, Handley seeks and speaks a language which recovers a spiritual relationship to the earth.

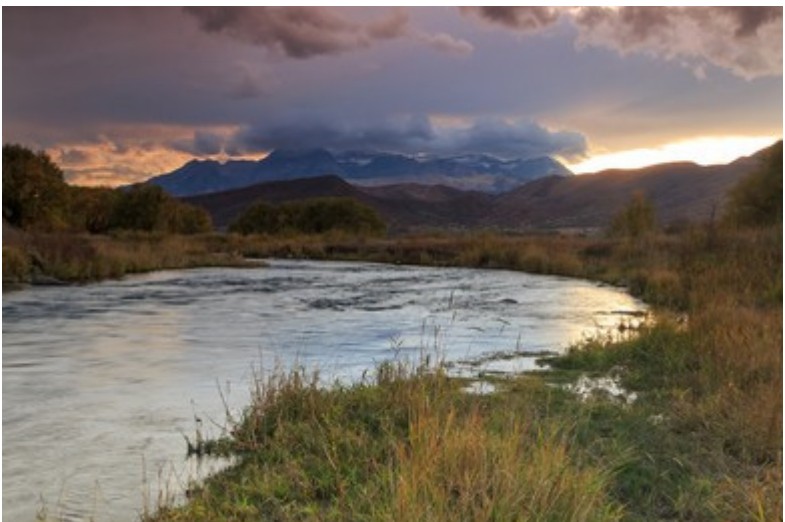

Provo River[25].

Tacey Atsitty and Farina King are both Indigenous Mormon women, scholars, and artists. Their work locates language—and mediums—through which to address the disconnections they are uniquely aware of. Tacey M. Atsitty, Diné (Navajo), is Tsénahabiłnii (Sleep Rock People) and born for Ta'neeszahnii (Tangle People).[26] She is a poet engaging disconnections between land, religion (particularly Mormonism), and culture. Her poem "Sunbeam" imagines the land through dream and the re-creation of poetry:

> Around noontime on Highway 666, we are driving to town. It is Pepper's fifth birthday. My dad is working. He is probably running laps with students. Cloudless. Our two vehicles leave the Chuskas. I want a sucker. Cheii takes me south. There are six of them in the other car; they turn north. It is too bright today. Two weeks ago my mom dreamt of night birds chanting amid juniper berries. Today, the land formations look like owls. I leave Little Water Trading Post with Minnie Mouse's heart in my mouth. Pepper is singing, "Jesus Wants Me for a Sunbeam," with our cousin-sister when—Mom was holding Baby in the front passenger seat and shot a look over to her sister—My little brother sips root beer while Baby sleeps. It was May. I sit alone in the back of my cheii's truck, wiping rouge across my eyelids. I don't understand the dream or the land—Grandma clenches my hand as we stand on the road, watching the sun take them.
>
> peppergrass gathered in a pink cup, here, Daddy. (Atsitty 2018)

Atsitty moves between dream, physical movement, and alterations of the land with the Mormon children's song "Jesus Wants Me for a Sunbeam" weaving through the motion of her words. She traverses worlds—night birds, juniper berries, land formation, the sun, peppergrass; and then suckers, Minnie Mouse, root beer, rouge, and a pink cup. The traversing meanders through religious territory: Atsitty's father was part of the Indian Placement program enacted by the Mormon church and formalized in 1954 (Torres 2018; Morgan 2009; Jacobs 2016; Murphy 2020) and the Mormon children's hymn she employs highlights a sense of both connection and disconnection. I look to Atsitty as a vibrant witness to Mormon theopoetics of creation: through poetry she engages the work of attending to disconnection and awakening to the earth. Importantly, poems such as "Sunbeam" draw attention to false dichotomies we have drawn between humans and the land, past and present. Her engagement bids attention and discussion: what do the night birds, juniper berries, land formation, sun, and peppergrass mean in the context of Indigenous history to Mormonism? What does "Sunbeam" say about Mormonism's current relationship to land? How does her dream fit into the broad Mormon imagination and dream of the future?

Farina King is "Bilagáanaa" (Euro-American), born for "Kinyaa'áanii" (the Towering House Clan) of the Diné (Navajo).[27] She is a scholar and an artist, gesturing to the earth through photography.

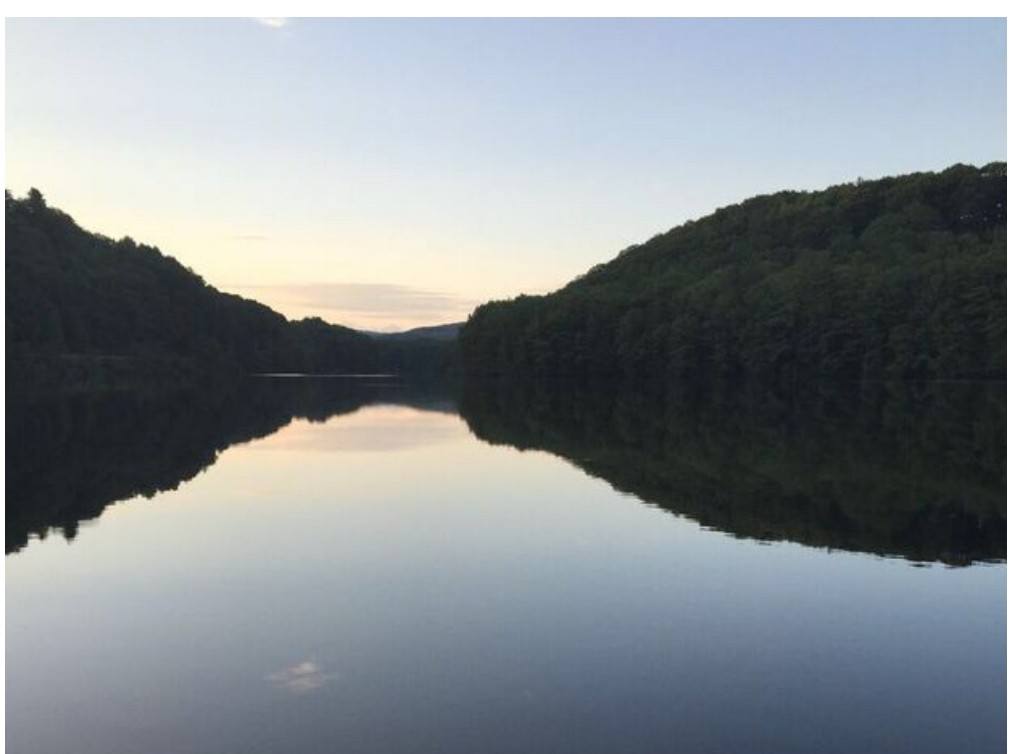

White River Junction, Vermont. Photo credit: Farina King, 2016.

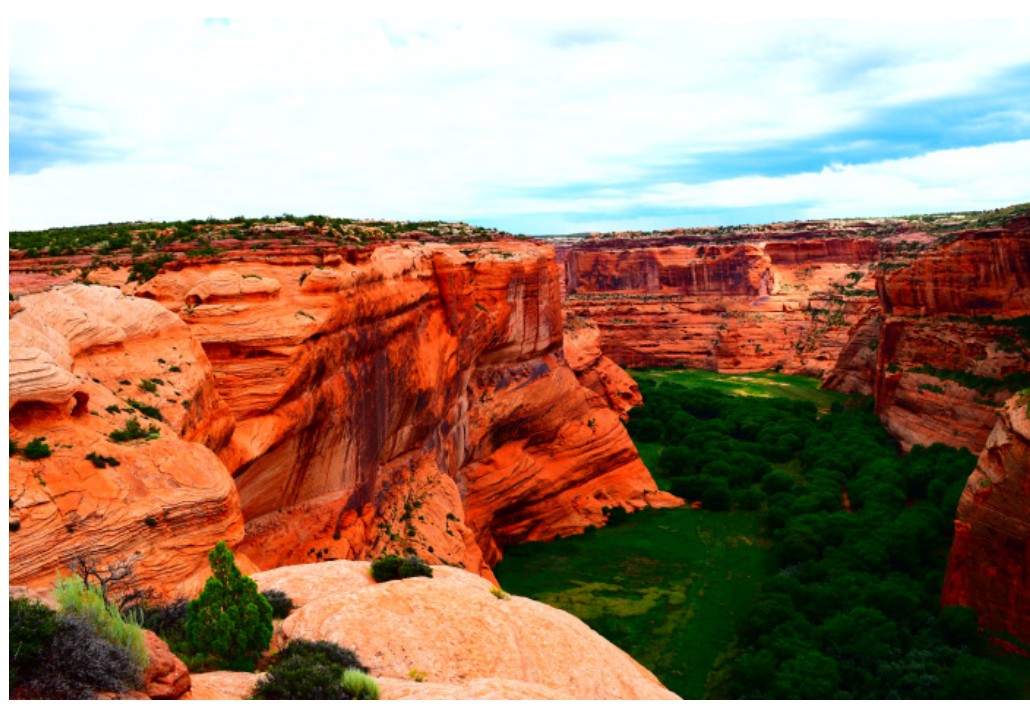

Canyon de Chelly. Photo credit: Farina King, 2016[28].

King's art captures the earth in the artistic language of respect and the reciprocity of beauty.[29] Her photography, meditating on the lands of her people, intentionally accompanies her professional scholarship focused on Indigenous Studies. Her unique intersections as a Mormon scholar and Indigenous woman find a certain voice in the theopoetic medium of photography, drawing attention to the land as she witnesses it. Her photos invite meditating, asking, and responding. For example, what does it mean that King includes photos of White River Junction, Canyon de Chelly, Monument Valley, Antelope Canyon, and more images from the Navajo Nation alongside her prestigious scholarship? What witness does King bring to the land, and what angle? What is visible in these photographs, and what is not? How does the medium of photography communicate King's particular witness?

Michiko Chiba is a Japanese woman in my church community. Her English is limited and my Japanese is nonexistent, so we communicate primarily through her garden. There, Michiko speaks softly to the rows of beans and peppers, and I try to understand. Often, her roommate tells me, Michiko will weep when she sees plants being mistreated. More than once, I have brought her my ailing houseplants for doctoring. Her porch is usually full of pots containing sick plants from various friends and neighbors, and she goes from one to another singing Japanese lullabies. Michiko understands little of the church services she attends every week, but she frequently comes to church with baskets of vegetables from her garden. *Take, eat*, she gestures. This is her language among us, and it is also her faith. The earth is alive, and she knows its language. Her garden knows her too, and I suspect the plants could tell me all the stories I long to know about this woman far from her first home, who shows me pictures of cherry trees in Japan, and prays on her knees in the garden.

What does it mean to communicate through the earth as a language? How might we look to the earth, as Michiko does, for communication, comfort, and faith?

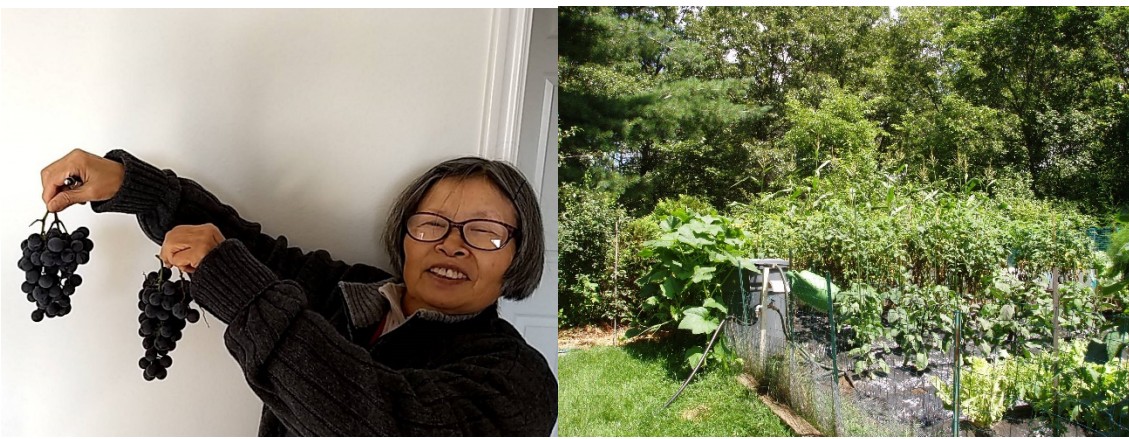

Michiko Chiba and her garden (Lexington, MA). 2020. Photos by Lori Parkinson. Used with permission.

Each of the examples I have given are theopoetic, gesturing to sites of theological reflection born of embodiment, place, and love in the honest languages of Mormon people. The Mormon body needs more theopoetic engagement to revive the communal imagination and remind tired running feet of their connectedness to the earth. Mormons need to remember the feeling of faith and listen to where and how it calls, to how it invites imagination and response anew. The Mormon theopoetics I have pointed to are sweet with imagination, with language intent on listening, hearing, and understanding. So it must be for those who share the vision, who seek to "blur the horizon between heaven and earth," and who hope to evangelize the word of life they read in this good earth. Rebecca Chopp argues that "the poetic can allow voices that were *always* present to finally be heard over the din of dominant discourse . . . . Employing non-propositional, aesthetic, and poetic dimensions of communication may help to bypass dominant cultural systems that might otherwise dismiss and/or filter out the content were it in another medium" (qtd in Keefe-Perry et al. 2014). For the messages of Mormon's potential environmental ethic to sink into the oceanic immensity of our beings Saints need to listen to more George Handleys, Tacey Atsittys, Farina Kings, and Michiko Chibas. Saints need the free imagination with no respect to persons.

By this I imagine the underrepresented wealth of women's voices and bodies, not necessarily from the pulpit but in poem, in song, in dance, attending to the earth in community.[30] How could a centralized tradition intent on authority receive and engage with George Handleys, Tacey Atsittys, Farina Kings, and Michiko Chibas? An answer to this question is the very wind of new life I hope for. To begin, I imagine calls for poetic reflections on place, so important to the Mormon story. Places like Michiko's Japan, and her garden; places like Rock Canyon and Canyon de Chelly. I imagine community scripture study groups focused on our relation to earth. I imagine wild church worship among friends and neighbors who seek to know the land more intimately and respond to her call. I imagine gathering in the places our bodies know and love and seeking communally, poetically, with our feet on the earth, our ethical imperative born of that love. I imagine a flourishing of Latter-day Saint Earth Stewardship groups.[31] I imagine communal prayers of lament and prayers of repentance, prayers of hope and prayers of commitment. I imagine an ongoing creation, a re-creation that "blurs the line between heaven and earth."

## 4. Conclusions

A theopoetic engagement with Mormonism's radically unique conception of creation, rooted in embodiment and place, is a spring of water in the wilderness to revive our dreary imagination. Restoration is the essential message of the Mormon story, promising "all things new," by bringing the good news of Jesus Christ out of obscurity with prophetic imagination. The Mormon narrative of creation typifies the prophetic imagination, imagining a subversive alternative to the dominant narrative of Christian cosmology. This

imagination is the heart of the Mormon story: imagining how the biblical origin story might have been lost (literally) in translation, and how it could emerge from the wilderness. The Mormon creation narrative embraces the chaos of unorganized and eternal matter, the *tehom* that so troubled the early patristic waters. Mormons instead assert creation as organization, a gathering of matter and spirit. The order of creation in this understanding is first spiritual, then physical; the physicality of material creation is the work and glory of our lives and our God. Belief and response must be intimately connected to sustain integrity, yet the Mormon story has seen deep and continuing disconnection between belief and response concerning the natural world. For many possible reasons, Mormon people moved away from the dangerously radical consequences of prophetic imagination in pastoral action, remaining bound to conservatism and hindering an ability to imagine the implications of a radical theological imagination.

This disconnect between Joseph Smith's prophetic imagination, the ethic implied in the story of creation, and current Mormon practice toward the environment, is a wound in the flesh of the collective body. This awful woundedness begs diverted attention, and I have proposed a return to imagination as a spiritual practice. I have danced with a few living examples which invite the kinds of dialogical questioning which can work as a healing salve to restore Mormon imaginative and responsive capacity. A theopoetic response to the Mormon story of creation is rooted in an embodied connection to the various places and contexts of this world. Theopoetics, arising first from connection and seeking diverse methods of expression, draws attention to that which is easily unsung and propels us back to our knees.

**Funding:** This research received no external funding.

**Institutional Review Board Statement:** Not Applicable.

**Informed Consent Statement:** Not Applicable.

**Data Availability Statement:** Not Applicable.

**Conflicts of Interest:** The author declares no conflict of interest.

## Notes

[1] The official and preferred name of the church is The Church of Jesus Christ of Latter-day Saints. I will use the colloquial 'Mormon,' 'Latter-day Saint,' or 'Saint' throughout for convenience and readability.

[2] I use this nickname to refer to the Church in addition to the well-known nickname 'Mormon'.

[3] This is developed in (Hubler 1955, p. 980).

[4] Augustine takes up creation ex nihilo in his *De Trinitate*. The problem of evil has its roots in the creation; for Augustine, the answer is in the being of God and that being's creation of the world. First, Augustine decisively unifies the trinity as one substance—"unchangeable, incorruptible, eternal, immortal, and infinite"—with no separation between the identities. Consequently, and intricately connected, he answers the problem of evil by saying that God's "incorruptible, eternal, immortal, and infinite" nature is wholly and completely responsible for all creation.

[5] I use male pronouns for God intentionally when discussing God's conception according to the early church fathers, who imagined God male. For the rest of my discussion, I use no pronouns for God.

[6] I use the word 'restoring' intentionally. Joseph Smith's prophetic mission was to restore the original gospel of Jesus Christ that had been changed since Jesus' day.

[7] Moses 3:5.

[8] The first stage of creation, according to Augustine, is a forming of matter. From that matter, all things are made. "Thus, creation *ex nihilo* is preserved without caving to the darksome deep." Quoted in Keller, *Face of the Deep*, 16.

[9] Several traditions claim the scripture of Joseph Smith as theologically relevant and employ them in different ways. This paper interrogates only The Church of Jesus Christ of Latter-day Saints.

[10] Clare (2009) uses this term in her master's thesis, "*Ex Materia* and the image of God: Imagining a Mormon Environmental Ethic." Master's Thesis: University of Mary Washington. She also traces the difference between *ex nihilo* and *ex materia* conceptions of creation.

[11] Joseph Smith taught, "The intelligence of spirits had no beginning, neither will it have an end. That is good logic. That which has a beginning may have an end. There never was a time when there were not spirits. . . . Intelligence is eternal and exists upon

a self-existent principle. It is a spirit from age to age and there is no creation about it. All the minds and spirits that God ever sent into the world are susceptible of enlargement" (quoting Joseph Smith sermon 7 April 1844).

12  D&C 93:29

13  Abraham 3:24.

14  Mormon doctrine is not monolithic, so it is tenuous to claim a single theological view. See for example Faulconer (2020), *Thinking Otherwise: Theological Explorations of Joseph Smith's Revelations*, Provo, Utah: Neal A. Maxwell Institute for Religious Scholarship. Teachings such as that of God organizing intelligences are found in church handbooks used to teach adult and youth classes.

15  Galli (2011), for example, postulates a Gaia hypothesis in relation to LDS scripture.

16  And others, like A. N. Whitehead, Charles Hartshorne, David Ray Griffin, and Willie Jennings.

17  Moses 7:28, see also Mormon scholars Givens and Givens (2017), *The God Who Weeps: How Mormonism Makes Sense of Life*.

18  Moses 1:39.

19  George Handley writes, "Deep in the visions and translations of the seer, Joseph Smith, even many Mormons have missed the implications of the belief that the new earth and new heaven would be this earth, this place here, now. The theology of such a restoration promises that the very stuff of our mortal lives will become the stuff of our heavenly existence . . . This is a philosophy of hope, hope that mundane, physical life, when properly cared for, might become the stuff of eternity." (Handley 2010, p. 121). Additionally, I am attending to dualism as treated in philosophy: continental philosophy in the school of phenomenology has explored the dangers of reductive dualism, and I accordingly discard a tendency toward spiritual and physical dualism: it is a false dichotomy here to separate the physical from the spiritual. See Maurice Merleau-Ponty and Donald A. Merleau-Ponty and Landes (2012), *Phenomenology of Perception*.

20  *The Book of Mormon: Another Testament of Jesus Christ* claims to be a record of Indigenous peoples on the American continent. Accordingly, regardless of the truth of this claim, Mormon believers are poised to be foremost in learning from their wisdom, in and out of the *Book of Mormon*.

21  In addition to Handley and others aforementioned, see (Williams et al. 1998; Rogers and Godfrey 2019; Anderson and Dubrasky 2021).

22  The varied history of Indigenous Mormons, for example, as well as the scholars and individuals I have briefly surveyed provide important exceptions.

23  I borrow this language from Faulconer (2020). *Thinking Otherwise: Theological Explorations of Joseph Smith's Revelations*.

24  Willie Jennings makes similar moves in *The Christian Imagination*, where he situates creation ex nihilo as part of a framework of dominance: "I want Christians to recognize the grotesque nature of a social performance of Christianity that imagines Christian identity floating above land, landscape, animals, place, and space." This imagination was constructed ex nihilo, just as ex nihilo doctrine itself was. (Jennings, *The Christian Imagination*, 293).

25  Image via shutterstock (https://www.shutterstock.com/search/provo+river) 21 April 2021.

26  See Dr. Atsitty's website, http://taceymatsitty.com/bio/ April 2021.

27  See Dr. King's website, https://farinaking.com/biography/ April 2021.

28  Photographs accessed via https://farinaking.com/photos/ in April 2021.

29  Kimmerer, *Braiding Sweetgrass.*

30  For example, Instagram accounts like Womb Sisters or the peripheral Mormon Women's magazine Exponent II.

31  See https://ldsearthstewardship.org/.

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
