# Peer review of "Disconnection and the Healing Practice of Imagination for Mormon Environmental Ethics"

_religions, doi:10.3390/rel12110948_

Round 1

Reviewer 1 Report

This is a beautifully written and deeply compelling intervention into the environmental implications of Latter-day Saint scripture and theology. It also draws on cutting-edge work by Latter-day Saint scholars and artists to set forth a rich suggestion about how the ostensible gap between Latter-day Saint doctrinal commitments and the lived practices of believing Latter-day Saints might be closed. Especially important is that this paper is deeply enmeshed with the best and most current thinking in the Latter-day Saint intellectual tradition---its leading theologians, most important environmental spokespeople, and its most compelling artists. On this score, this is exemplary work.

There are nevertheless a couple of concerns I have about the way the paper goes about its work.

First, I think the author might be a little more careful about describing what's normative within the Latter-day Saint tradition. For one, there are various religious groups claiming the scriptural heritage derived from Joseph Smith, but only one of these is discussed seriously. Perhaps more importantly, however, the authors speaks of "Mormon doctrine" as if this were a stable and readily identifiable thing, and this is questionable (and has been questioned in important publications). It would be better, in my view, if the author were to speak of what Latter-day Saint scripture commits its adherents to, and of what the Latter-day Saint tradition has tended to endorse theologically---rather than of anything that might sound like a monolithic "Mormon doctrine."

Second, the author sometimes seems a little unsure of the audience of the paper. At times, the author clearly assumes a broad readership in religious studies, clarifying terms for readers and acquainting them with the tradition. At other times, however, the author seems to assume that this piece is an insider essay, written by a Latter-day Saint for other Latter-day Saints. When "we" talk appears ("we need to do X"), for instance, or when Joseph Smith becomes "Joseph" rather than "Smith," the author begins to exclude non-Latter-day Saint readers from the discussion. If the primary aim of the essay is to address Latter-day Saints directly with a plea, there is probably a better venue for it than a journal like Religions. If the aim is to speak to a larger religious studies audience, however, some editing work needs to be done to bring the tone into coherence and to keep the primary audience always in mind.

Third, the author ought, in my view, to address one major objection that might be raised about the argument in this paper. There is real promise in theopoetics, and the argument set forth in the paper is compelling. The Latter-day Saint tradition, especially in the majority-institution of The Church of Jesus Christ of Latter-day Saints, is, however, profoundly centralized, such that the majority of believing adherents don't look to artists for guidance in understanding. How might theopoetics make for a real healing of the gap the author identifies when confronted with this obstacle? Are there ways to get the photography of Farina King, for instance, into the hands of average believers? Are there ways to get the writings of George Handley, to take another example, before the eyes of non-scholars? How would the author respond to this challenge?

These concerns are, I think, all easily dealt with through minor revision. I'd encourage the author to take them seriously while revising the paper for publication, which I highly recommend.

Reviewer 2 Report

I love this essay. Two particular strengths stand out: the theological articulation of the potentiality of Mormon notions of creation for an applied ecological ethic and the engagement with the theopoetics of Indigenous Mormon women. The article is thoughtful, cogently argued, and well-written. 

There are a couple of significant concerns that detract form the overall effectiveness of the article. The evidence presented for the lack of a Mormon environmental ethic is all from the twenty-first century and does not appear to allow for variation over time or for more nuance in actual practice, particularly among Indigenous Mormon women. In my experience, growing up male and Mormon in the 1970s and 80s in southern Idaho with Indigenous heritage, an environmentally oriented service ethic was a central part of my Latter-day Saint experience, particularly through the Boy Scouts of America. Earlier generations of Mormonism produced notable environmentalists such as Terry Tempest Williams, Stewart Udall, and Hugh Nibley. Within Indigenous communities, a particular focus of the article, notable exceptions to the trend claimed by the author include Helen Sekaquaptewa (Hopi), Helen Yellowman (Navajo), Ella Bedonie (Navajo), and P. Jane Hafen (Taos Pueblo). 

This overstatement of the problem of the lack of a Mormon environmental ethic can be easily addressed. First, the author might limit claims of a lack of Mormon environmental ethic to the twenty-first century or provide evidence for a longer historical trend. Second, the author can more explicitly acknowledge the diversity within Mormonism both politically and ethnically from the outset. Third, the author can at least acknowledge and cite more broadly the existing literature on Mormon environmentalism: consider New Genesis: A Mormon Reader on Land and Community, edited by Terry Tempest Williams, William B. Smart, and Gibbs M. Smith (1998); The Earth Will Appear as the Garden of Eden: Essays on Mormon Environmental History, edited by Jedidiah Rogers and Matthew Godfrey (U of Utah Press, 2019); and Blossom as the Cliffrose: Mormon Legacies and the Beckoning Wild, edited by Karin Anderson and Danielle Dubrasky (Torrey, 2021). The work of Indigenous Mormon women might be more fairly represented by citing and considering the following: the essay by P. Jane Hafen in New Genesis, poetry by Tacey Atsitty, Farina King, and Stacie Denetsosie in Blossom as the Cliffrose; Me and Mine: The Life Story of Helen Sekaquaptewa as told to Louise Udall (U of AZ Press, 1969); Beyond the Four Corners of the World: A Navajo Woman's Journey [about Ella Bedonie] by Emily Benedek (U of OK Press, 1995), and the documentary film Into America: The Ancestor's Lands about Helen Yellowman produced by her grandson Angelo Baca and Nadine Zacharias (2012); Decolonizing Mormonism: Approaching a Postcolonial Zion, edited by Gina Colvin and Joanna Brooks (U of UT Press, 2018); and Essays on American Indian and Mormon History, edited by P. Jane Hafen and Brenden Rensink (U of UT Press, 2019). 

A couple of minor factual errors need correction. 1) Footnote 45 "is a record" should be changed to "claims to be a record." Also, it would be helpful given the article's focus on Indigenous women to cite Dakota Latter-day Saint Elise Boxer's critique of the historical claims of the Book of Mormon in her essay, "The Book of Mormon as Mormon Settler Colonialism" in Essays on American Indian & Mormon History. 2) Line 424 claims that the Mormon Indian Student Placement Program "began" in 1954. A more accurate statement would note that the LDS Relief Society formalized the ISPP as an official church program in 1954.  Informal and extralegal removals of Indigenous children and placement in Mormon homes began more than a century earlier (see Margaret Jacobs, "Entangled Histories: The Mormon Church and Indigenous Child Removal from 1850 to 2000," Journal of Mormon History [2016] 42.2, 27-60 and Thomas Murphy, "Views from Turtle Island: Settler Colonialism and Indigenous Mormon Entanglements," in The Palgrave Handbook of Mormonism [2020], 751-779).

Also, the Bibliography fails to cite any of Tacey Atsitty's or Farina King's scholarship despite the central focus on them in the second half of the essay. Please provide full citations for their poetry and scholarship in the Bibliography and consider including additional pieces beyond those discussed in the article (such as those in Blossom as the Cliffrose). 

Finally, please consider editing the abstract to better reflect the article's engagement with the theopoetics of Indigenous Mormon women. 

Reviewer 3 Report

This is a refreshing paper because it is written by an insider who understands the Church of Jesus Christ of Latter-day Saints, both in doctrine, as well as in practice. The author approaches the culturalisms within the church respectfully, but also critically. 

The lit review succeeds in placing the Latter-day Saint Church within a greater discussion of theological environmentalism. I am confident that few understand the church's genesis story, and to the extent that it distinguishes itself because of its scripture taken from the Pearl of Great Price. This was a strong point in this paper. 

The author also chooses to insert themself into the narrative, and while risky, it seemed to work well. Alongside Atsitty, King, and Chiba, the author uses themself as an example of the theopoetic, particularly in their description of rock canyon. 

Well done.

Author Response

Thank you for your engagement!